

# Inferring the instability of a dynamical system from the skill of data assimilation exercises

Yumeng Chen[1], Alberto Carrassi[1,2,3], and Valerio Lucarini[3,4]

[1]Department of Meteorology and NCEO, University of Reading, UK
[2]Mathematical Institute, University of Utrecht, NL
[3]Centre for the Mathematics of Planet Earth, University of Reading, UK
[4]Department of Mathematics and Statistics, University of Reading, UK

**Correspondence:** Yumeng Chen (yumeng.chen@reading.ac.uk)

**Abstract.** Data assimilation (DA) aims at optimally merging observational data and model outputs to create a coherent statistical and dynamical picture of the system under investigation. Indeed, DA aims at minimizing the effect of observational and model error, and at distilling the correct ingredients of its dynamics. DA is of critical importance for the analysis of systems featuring sensitive dependence on the initial conditions, as chaos wins over any finitely accurate knowledge of the state of the system, even in absence of model error. Clearly, the skill of DA is guided by the properties of dynamical system under investigation, as merging optimally observational data and model outputs is harder when strong instabilities are present. In this paper we reverse the usual angle on the problem and show that it is indeed possible to use the skill of DA to infer some basic properties of the tangent space of the system, which may be hard to compute in very high-dimensional systems. Here, we focus our attention on the first Lyapunov exponent and the Kolmogorov-Sinai entropy, and perform numerical experiments on the Vissio-Lucarini 2020 model, a recently proposed generalisation of the Lorenz 1996 model that is able to describe in a simple yet meaningful way the interplay between dynamical and thermodynamical variables.

## 1 Introduction

### 1.1 Lyapunov vectors and related measures of chaos in a nutshell

The dynamics of the atmosphere or of the ocean are characterised by chaotic conditions, which, roughly speaking describes the property that a system has sensitivity to initial states. This means that, even in the presence of a perfect model, small errors in the initial conditions will grow in size with time, until the forecast becomes *de facto* useless (Kalnay, 2002)[1]. A mathematically-sound technique for studying the sensitivity to initial conditions of a system amounts to studying the properties of its tangent space. In particular, under fairly general mathematical conditions for a deterministic $n-$dimensional system whose asymptotic dynamics takes place in a compact attractor, one can define $N$ Lyapunov exponents (LEs) $\lambda_1, \ldots, \lambda_N$, which are the asymptotic rates of amplification or decay of infinitesimally small perturbations with respect to a reference trajectory. Usually, the LEs are

---

[1]In the words of Ed Lorenz: "Chaos: When the present determines the future, but the approximate present does not approximately determine the future", see https://tinyurl.com/faf3pnda.





ordered according to their value, with $\lambda_1$ being the largest. Unless the system feature symmetries, all the LEs are distinct, and in the case of continuous time dynamics, one of them vanishes corresponding to the direction of the flow and defining the neutral tangent space. Once ordered from the largest to the smallest, the sum of the first $k$ LEs gives the asymptotic growth rate of a $k-$volume element defined by $k$ displaced infinitesimally nearby the reference trajectory plus the reference trajectory itself.

Additionally, if $\lambda_{n_0}$ denotes the smallest non-negative LE, in many practical applications one can estimate the Kolmogorov-Sinai (or metric) entropy $\sigma_{KS}$, which defines the rate of creation of information of the system due to its instabilities, can be identified with $\sum_{i=1}^{n_0} \lambda_i$ (Pesin's identity). Finally, it is possible to use the spectrum of LEs to define a notion of dimension for the attractor of a chaotic systems. The Kaplan-Yorke conjecture, which follows from the estimate of the rate of growth of the infinitesimal $k-$volume, indicates that the information dimension of a chaotic attractor is given by $D_{KY} = p + \sum_{i=1}^{p} \lambda_i / |\lambda_{p+1}|$,

where $p$ is the largest index such that $\sum_{i=1}^{p} \lambda_i \geq 0$. In systems where the phase space contracts (the large class of dissipative systems), one has $D_{KY} < n$. Roughly speaking, larger values of $\lambda_1$, of $\sigma_{KS}$ and of $D_{KY}$ are associated with conditions of high instability and low predictability for the flow. This is clearly an extremely informal presentation of some of the features and properties of the LE; see Eckmann and Ruelle (1985) for a now-classic discussion of these topics.

It is indeed possible to associate each LE with a physical mode. Ruelle (1979) proposed the idea of performing a covariant

splitting of the tangent linear space such that the basis vectors are actual trajectories of linear perturbations. The average growth rate of each of the covariant Lyapunov vector (CLVs) equals one of the LE. This idea was first implemented by Trevisan and Pancotti (1998) for studying the properties of the Lorenz 1963 model (Lorenz, 1963). Separate algorithms for the computation of CLVs were proposed in Ginelli et al. (2007) and Wolfe and Samelson (2007); see the recent comprehensive review by Froyland et al. (2013). Note that the CLVs corresponding to the positive (negative) LEs span the unstable (stable) tangent

space.

Recently, Lyapunov analysis of the tangent space was the subject of a special issue edited by Cencini and Ginelli (2013) and the book by Pikovsky and Politi (2016). Detailed Lyapunov analyses of geophysical flows on models of various levels of complexity have been recently reported (e.g., Schubert and Lucarini, 2015; Vannitsem and Lucarini, 2016b; Vannitsem, 2017; De Cruz et al., 2018b). Note that in many applications one might want to alter some of the aspects of the classical Lyapunov

analysis by accommodating for the study of how instabilities grow on a finite-time horizon (Palmer and Zanna, 2013) or of the growth of finite-size errors (Cencini and Vulpiani, 2013). In many applications, such non-asymptotic measures of error growth are more relevant than the classic Lyapunov analysis.

## 1.2   Data assimilation in chaotic systems: the signature and the use of chaos

Data assimilation (DA; Asch et al., 2016) refers to the family of theoretical and numerical methods that optimally combines

data with a dynamical model with the goal of improving the understanding of the phenomenon under study, enhancing the prediction skill, and quantifying the associated uncertainty. Data assimilation has long been studied and developed in the geosciences. It is an unavoidable piece of the operational numerical weather prediction workflow but it is nowadays used in a growing range of diverse areas of science (Carrassi et al., 2018).





Numerical evidences and recent analytical proofs have shown that, under certain conditions of the observations (their types, spatio-temporal distribution, and accuracy), the performance of DA with chaotic dynamics relates directly to the instability properties of the dynamical model where data are assimilated. One can thus in principle use the knowledge of the dynamical features to inform not only the design of the DA that better suits the specific application - *e.g.*, how many model realizations for the Monte Carlo based DA methods, the length of the assimilation window in variational DA - but also the best-possible observational deployment.

A stream of research has shed light on the mechanisms driving the response of the ensemble-based DA (Evensen, 2009), *i.e.* its functioning and suitability, when applied to chaotic systems. A recent comprehensive review can be found in Carrassi et al. (2021), while we succinctly recall the main findings in the following. In the deterministic linear and Gaussian case with Kalman filter (KF) and smoother (KS), it has been analytically proved that the error covariance matrices converge in time onto the model's unstable–neutral subspace, *i.e.* the span of the backward Lyapunov vectors (BLVs), or of the covariant Lyapunov vectors (CLVs), associated to the non-negative Lyapunov exponents (LEs, Bocquet et al., 2017; Bocquet and Carrassi, 2017). These results have then been shown numerically to hold for the ensemble Kalman filter/smoother in weakly nonlinear regimes (EnKF/EnKS; Evensen, 2009) by Bocquet and Carrassi (2017). In practice, for sufficiently well observed scenarios, the error of the state estimate is fully confined within the unstable-neutral subspace. Because this subspace is usually much smaller than the full system's phase space, the above convergence results imply that an ensemble size as large as the unstable-neutral subspace dimension suffices to achieve satisfactorily performance, *i.e.* to track the "true" and effectively reduce the estimation error thus leading to a substantial computational saving.

The picture slightly changes in the presence of a degenerate spectrum of LEs. This degeneracy often arises in systems with multiple scales or in coupled dynamics (Vannitsem and Lucarini, 2016a; De Cruz et al., 2018a): the degeneracy usually regards the unstable-neutral portion of the LE spectrum. In these cases it is necessary to increase the ensemble size to account for all of the degenerate modes (Tondeur et al., 2020; Carrassi et al., 2021).

The necessity for going beyond the number of asymptotic unstable-neutral modes is also connected to the local variability of the instantaneous instabilities along a system's trajectory. A recent study performed on a quasi-geostrophic model of the atmosphere (Lucarini and Gritsun, 2020) provided a strong evidence that the large heterogeneity of the atmospherics's predictability is due to the presence of substantial variability in the number of unstable dimensions (Lai, 1999) of the unstable periodic orbits (UPOs) populating the attractor and defining the skeletal dynamics of the system (Auerbach et al., 1987). As a result of the fact that the orbit of a chaotic system shadows the UPOs supported on the attractor in some of its regions , certain directions of the stable space experience finite-time error growth due to locally important instabilities, causing the need for a larger ensemble size than the dimension of unstable-neutral.

In the stochastic scenario, the above results still hold, although with some qualifications. Stochasticity usually injects noise in the system irrespective of the flow-dependent modes of instabilities. Consequently, and with a non-zero probability, it also injects error onto stable directions that would not have been otherwise influential in the long term. The trade-off between the frequency of the noise injection and its amplitude on the one hand, and the dissipation rate of stable modes on the other, determines the amplitude of the long term error along stable modes (Grudzien et al., 2018a). This mechanism implies the need





to include additional members in the ensemble to encompass weakly stable modes that experience instantaneous growth and,
although exacerbated by the presence of system noise, is present in any reduced-rank KFs as explained by Grudzien et al.
(2018b).

Moving away from the Gaussian and weakly-nonlinear scenario, the impact of instabilities on the functioning and performance of nonlinear DA, in particular particle filters (PFs, see *e.g.* Van Leeuwen et al., 2019) has also recently been clarified. Carrassi et al. (2021) have shown that the number of particles needed to reach convergence depends on the size of the unstable-
neutral subspace rather than the observation vector size.

We have seen how the knowledge of the LEs and LVs can be used to operate key choices in the implementation of ensemble-based DA schemes aimed at enhancing accuracy with the smallest possible computational cost. This view angle is made explicit in a class of DA algorithms that operates a reduction in the dimension of the model (*e.g.*, the assimilation in the unstable subspace, AUS, Palatella et al., 2013), of the data (Maclean and Van Vleck, 2021) or both (Albarakati et al., 2021).

### 1.3 This paper: data assimilation as a tool to interrogate the dynamics

While extremely appealing from a theoretical view-point and practically useful in low-to-moderate dimensional problems, the use of the dynamically informed DA approaches is difficult in high dimensions, where even just computing the asymptotic spectrum of LEs, let alone the very relevant state-dependent local LEs (LLEs), is very difficult or just impossible. As pointed out in Carrassi et al. (2021), the recent advent of powerful machine learning methods may open the path to efficiently emulate
the otherwise very costly recursive procedure to compute the LEs and vectors.

In this work, we attempt to reverse the view angle by posing the question: can we use the results of DA to infer some fundamental quantities of the underlying dynamics, such as the spectrum of the LEs or the Kolmogorov-Sinai entropy ($\sigma_{KS}$), that are, as aforementioned, difficult to compute in high-dimensions.

The paper is structured as the following: in Sect. 2, an upper bound of the root mean squared error of the Kalman filter for
the linear dynamics in the asymptotic limit is derived. In Sect. 3 we present the Vissio and Lucarini (2020) (VL20) model and its DA setup. The VL 2020 model is a recently proposed generalisation of the Lorenz (1996) model that is able to describe in a simple yet meaningful way the interplay between dynamical and thermodynamical variables. Additionally, the presence of qualitatively distinct set of spatially extended variables allows one to consider non-trivial cases of partial observations for DA exercises. Sect. 4 presents the main results of the paper by comparing the skill of the performed DA exercises with some
fundamental measures of instability of the VL20 model. Finally, in Sect. 5 we discuss our results and present perspectives for future investigations.

## 2 Kalman filter error bounds and Lyapunov spectrum

We are interested in searching for a further relation between the skill of EnKF-like methods applied to perfect (no model error) chaotic dynamics and the spectrum of LEs. We shall build our derivations on the results mentioned in Sect. 1.2 and reviewed in
Carrassi et al. (2021). In this section we set ourselves in a linear and Gaussian context, whereby the Kalman filter (KF) yields





the exact solution of the Gaussian estimation problem. Linear results will guide the interpretation of the findings in the general nonlinear setting with the EnKF.

At time $t_k$, let $\mathbf{x}_k \in \mathbb{R}^n$ and $\mathbf{y}_k \in \mathbb{R}^d$ be the state and observation vector, respectively. The (linear) model dynamics $\mathbf{M}_k \in \mathbb{R}^{n \times n}$ and observation model $\mathbf{H}_k \in \mathbb{R}^{p \times n}$ read

$$\mathbf{x}_k = \mathbf{M}_k \mathbf{x}_{k-1}, \tag{1}$$

$$\mathbf{y}_k = \mathbf{H}_k \mathbf{x}_k + \mathbf{v}_k. \tag{2}$$

The observation noise, $\mathbf{v}_k$, is assumed to be a zero-mean Gaussian white sequence with statistics

$$\mathrm{E}[\mathbf{v}_k \mathbf{v}_l^\top] = \delta_{k,l} \mathbf{R}_k, \tag{3}$$

with $\mathrm{E}[\,]$ being the expectation operator, $\delta_{k,l}$ the Kronecker's delta function, and $\mathbf{R}_k$ the error covariance matrix of the observa-
tions at time $t_k$. For the sake of notation clarity, we assume that the model dynamics is non-degenerate so that all its Lyapunov exponents are distinct; we note that the extension to the general degenerate case is possible.

The singular vector decomposition (SVD) of the model dynamics between $t_k > t_l$ reads:

$$\mathbf{M}_{k:l} = \mathbf{U}_{k:l} \mathbf{\Lambda}_{k:l} \mathbf{V}_{k:l}^\mathrm{T}, \tag{4}$$

where $\mathbf{U}_{k:l}$ and $\mathbf{V}_{k:l}$ are non-degenerate orthogonal matrices and $\mathbf{\Lambda}_{k:l}$ the diagonal matrix of singular values. For $t_l \to -\infty$
the left singular vectors, $\mathbf{U}_{k:l}$, converge to the backward Lyapunov vectors (BLVs) at $t_k$, and, similarly for $t_k \to \infty$ the right singular vectors, $\mathbf{V}_{k:l}$, converge to the forward Lyapunov vectors (FLVs) at $t_l$. The singular values (SVs) in $\mathbf{\Lambda}_{k:l}$ converge to $n$ distinct values of the form $\mathbf{diag}(\mathbf{\Lambda}_{k:l})_i = \exp(\lambda_i(t_k - t_l))$, in which $\lambda_i$ are the Lyapunov exponents (LEs) in descending order, $\lambda_1 > \lambda_2 > \ldots \lambda_{n_0} = 0 > \lambda_{n-1} > \lambda_n$. The $n_0$ non-negative LEs identifies the $n_0$ unstable-neutral modes.

Let define the *information matrix*:

$$\mathbf{\Gamma}_k = \sum_{l=0}^{k-1} \mathbf{M}_{k:l}^{-\mathrm{T}} \mathbf{H}_l^\mathrm{T} \mathbf{R}_l^{-1} \mathbf{H}_l \mathbf{M}_{k:l}^{-1} = \sum_{l=0}^{k-1} \mathbf{M}_{k:l}^{-\mathrm{T}} \mathbf{\Omega}_l \mathbf{M}_{k:l}^{-1}, \tag{5}$$

which measures the "observability" of the state at $t_k$, with $\mathbf{\Omega}_l = \mathbf{H}_l^\mathrm{T} \mathbf{R}_l^{-1} \mathbf{H}_l$ being the *precision matrix* of the observations mapped to the model space. Moreover, let $\mathbf{U}_{+,k}^\mathrm{T}$ be a matrix whose columns are the $n_0$ unstable and neutral BLVs of the dynamics $\mathbf{M}_k$. Bocquet and Carrassi (2017) have shown that, if the following three conditions hold, **(i)** the unstable-neutral modes are sufficiently observed, such that

$$\mathbf{U}_{+,k}^\mathrm{T} \mathbf{\Gamma}_k \mathbf{U}_{+,k} > \epsilon \mathbf{I}_n \quad \epsilon > 0, \tag{6}$$

with $\mathbf{I}_n \in \mathbb{R}^n$ being the identity matrix, **(ii)**, the neutral modes, $\mathbf{u}$, are subject to the stronger constraint,

$$\liminf_{k \to \infty} \mathbf{u}_k^\mathrm{T} \mathbf{\Gamma}_k \mathbf{u}_k = \infty, \tag{7}$$





and, **(iii)** confining the initial error covariance matrix to the space of FLVs at time $t_0$, then the KF forecast error covariance matrix, $\mathbf{P}_k^f$, converges asymptotically to the sequence

$$\mathbf{P}_k = \mathbf{U}_{+,k}(\mathbf{U}_{+,k}^T \mathbf{\Gamma}_k \mathbf{U}_{+,k})^{-1} \mathbf{U}_{+,k}^T. \tag{8}$$

In real applications, the convergence (within numerical accuracy) occurs in long-but-finite times (Bocquet et al., 2017).

The asymptotic mean squared error of the forecast (MSEF) of the KF solution is given by the trace of Eq. (8),

$$n\text{MSEF} = \text{Tr}(\mathbf{P}_k) = \text{Tr}[\mathbf{U}_{+,k}(\mathbf{U}_{+,k}^T \mathbf{\Gamma}_k \mathbf{U}_{+,k})^{-1} \mathbf{U}_{+,k}^T] \tag{9}$$

$$= \text{Tr}[(\mathbf{U}_{+,k}^T \mathbf{\Gamma}_k \mathbf{U}_{+,k})^{-1}], \tag{10}$$

where, for last equality, we made use of the cyclic property of the matrix trace and the orthogonal relation of the BLVs, $\mathbf{U}_{+,k}^T \mathbf{U}_{+,k} = \mathbf{I}_{n_0}$. Equation (9) puts in evidence that the asymptotic MSEF depends on the observation constraint through the information matrix (data accuracy, encapsulated in $\mathbf{R}$, while data type and deployment encapsulated in $\mathbf{H}$) , but also on the unstable-neutral BLVs. Despite this, it is particularly involved to use Eq. (9) to derive a direct relation between the MSEF and the spectrum of LEs. This is because $\mathbf{U}_{+,k}$ is not invertible in general, and because one needs to make specific (often overly simplified) assumptions on the model dynamics and observations, *i.e.* on $\mathbf{M}_{k:l}$, $\mathbf{H}_l$ and $\mathbf{R}_l$, in order to get a treatable expression of the information matrix. In alternative, rather than a direct relation, we shall seek for informative bounds for the MSEF in terms of the LEs.

Let substitute the SVD of $\mathbf{M}_{k:l}$, Eq. (4), in the information matrix,

$$\mathbf{\Gamma}_k = \sum_{l=0}^{k-1} \mathbf{U}_{k:l}^{-T} \mathbf{\Lambda}_{k:l}^{-1} \mathbf{V}_{k:l}^{-1} \mathbf{\Omega}_l \mathbf{V}_{k:l}^{-T} \mathbf{\Lambda}_{k:l}^{-1} \mathbf{U}_{k:l}^{-1}. \tag{11}$$

For every $t_l$, the individual terms in the summation can be written as:

$$[\mathbf{U}_{k:l} \mathbf{\Lambda}_{k:l} \mathbf{V}_{k:l}^T \mathbf{\Omega}_l^{-1} \mathbf{V}_{k:l} \mathbf{\Lambda}_{k:l} \mathbf{U}_{k:l}^T]^{-1}. \tag{12}$$

We now define the maximum projection of the precision matrix onto the FLVs as:

$$\beta_l = \max_{\mathbf{h} \in \text{Im}(\mathbf{V}_{k:l}), \|\mathbf{h}\|=1} \mathbf{h}^T \mathbf{\Omega}_l^{-1} \mathbf{h}, \tag{13}$$

with $\|.\|$ being the Euclidean norm, and use it to get an upper bound for the inverse of each term, Eq. (12), in the information matrix summation,

$$\mathbf{U}_{k:l} \mathbf{\Lambda}_{k:l} \mathbf{V}_{k:l}^T \mathbf{\Omega}_l^{-1} \mathbf{V}_{k:l} \mathbf{\Lambda}_{k:l} \mathbf{U}_{k:l}^T \leq \beta_l \mathbf{U}_{k:l} \mathbf{\Lambda}_{k:l}^2 \mathbf{U}_{k:l}^T. \tag{14}$$

The inequality is based on the Löwner partial ordering of $\mathbb{R}^{n \times n}$ (*i.e.* the partial order defined by the convex cone of positive semi-definite matrices; see *e.g.*, Bocquet et al. (2017) their Appendix B). We shall use this partial ordering in the following derivations.





By defining the maximum of $\beta_l$ across all $0 \le t_l \le t_{k-1}$ as

$$\beta_k = \max_{l=0,\ldots,k-1} \beta_l, \tag{15}$$

we get the following lower bound for the information matrix:

$$\beta_k^{-1} \sum_{l=0}^{k-1} (\mathbf{U}_{k:l} \mathbf{\Lambda}_{k:l}^2 \mathbf{U}_{k:l}^{\mathrm{T}})^{-1} \le \sum_{l=0}^{k-1} [\mathbf{U}_{k:l} \mathbf{\Lambda}_{k:l} \mathbf{V}_{k:l}^{\mathrm{T}} \mathbf{\Omega}_l^{-1} \mathbf{V}_{k:l} \mathbf{\Lambda}_{k:l} \mathbf{U}_{k:l}^{\mathrm{T}}]^{-1} \tag{16}$$

$$= \mathbf{\Gamma}_k. \tag{17}$$

The bound reflects the effect of assimilating observations (the rhs) compared to the unconstrained free model run (the lhs) - note that $\mathbf{U}_{k:l} \mathbf{\Lambda}_{k:l}^2 \mathbf{U}_{k:l}^{\mathrm{T}} = \mathbf{M}_{k:l} \mathbf{M}_{k:l}^{\mathrm{T}}$.

Given that $\mathbf{U}_{k:l} \mathbf{\Lambda}_{k:l}^2 \mathbf{U}_{k:l}^{\mathrm{T}}$ is symmetric positive definite, we can invoke the aforementioned partial ordering for this class of matrices, and further develop the lower bound of the information matrix as

$$\mathbf{U}_{k:l} \mathbf{\Lambda}_{k:l}^2 \mathbf{U}_{k:l}^{\mathrm{T}} \le e^{2\lambda_1(t_k - t_l)} \mathbf{I} = \mathbf{D}_l, \tag{18}$$

where $e^{2\lambda_1(t_k - t_l)}$ is the largest eigenvalue of $\mathbf{U}_{k:l} \mathbf{\Lambda}_{k:l}^2 \mathbf{U}_{k:l}^{\mathrm{T}}$. The lower bound of the information matrix in Eq. (16) then becomes

$$\beta_k^{-1} \sum_{l=0}^{k-1} \mathbf{D}_l^{-1} \le \mathbf{\Gamma}_k. \tag{19}$$

Under the assumption that the assimilation cylcle is uniform in time, *e.g.* $\Delta t = t_k - t_{k-1} = t_{k-1} - t_{k-2} = \cdots = t_1 - t_0$, the summation of the diagonal matrices $\mathbf{D}_l^{-1}$ coincides with a geometric series with known sums:

$$s_i^{-1} = \sum_{l=0}^{k-1} \mathbf{D}_{l,ii}^{-1} = \begin{cases} \dfrac{e^{-2\lambda_1 \Delta t} - e^{-2\lambda_1(k+1)\Delta t}}{1 - e^{-2\lambda_1 \Delta t}} & \lambda_1 > 0 \\ k & \lambda_1 = 0 \end{cases}. \tag{20}$$

By using the lower bound, Eq. (19), and the orthogonality of the BLVs, $\mathbf{U}_{+,k}^{\mathrm{T}} \mathbf{U}_{+,k} = \mathbf{I}_{n_0}$, we get a lower bound for the information matrix projected onto the unstable-neutral subspace:

$$\beta_k^{-1} s_i^{-1} \mathbf{I}_{n_0} = \beta_k^{-1} s_i^{-1} \mathbf{U}_{+,k}^{\mathrm{T}} \mathbf{I}_n \mathbf{U}_{+,k} \le \mathbf{U}_{+,k}^{\mathrm{T}} \mathbf{\Gamma}_k \mathbf{U}_{+,k}. \tag{21}$$

We can thus finally use Eq. (21) in the expression of the MSEF, Eq. (9), and derive the following upper bound:

$$n\mathrm{MSEF} = \mathrm{Tr}[(\mathbf{U}_{+,k}^{\mathrm{T}} \mathbf{\Gamma}_k \mathbf{U}_{+,k})^{-1}] \tag{22}$$

$$\le \mathrm{Tr}(\beta_k s_i \mathbf{I}_{n_0}) \tag{23}$$

$$= \beta_k \sum_{i=1}^{n_0} \frac{1 - e^{-2\lambda_1 \Delta t}}{e^{-2\lambda_1 \Delta t} - e^{-2\lambda_1(k+1)\Delta t}} \tag{24}$$

$$\rightarrow \beta_k \sum_{i=1}^{n} \frac{1 - e^{-2\lambda_1 \Delta t}}{e^{-2\lambda_1 \Delta t}} \quad k \rightarrow \infty \tag{25}$$

$$= \beta_k n_0 (e^{2\lambda_1 \Delta t} - 1). \tag{26}$$





This upper bound incorporates the key players shaping the relation between the KF estimation error and the model dynamics. The presence of $\beta_k$ and $\Delta t$ reflects the observation modulation of the MSEF: the stronger the data constraint the smaller $\beta_k$ and $\Delta t$. The signatures of the model instabilities are in the term $n_0$, the size of the unstable-neutral subspace, and in $\lambda_1$, the error growth rate along the leading mode of instability, both related directly to the amplitude of the bound. Under a Bayesian interpretation, the factor $\beta_k$ can be seen as the likelihood of data, and the remaining terms in the bound altogether as the prior

distribution. Note that, if the dynamical model is stable (and independently of the data), $\lambda_1 = 0$, $s_i = \frac{1}{k}$ and the MSEF goes to zero asymptotically.

As alluded at the beginning of the section, a direct expression (*e.g.* an equality in place of a bound) relating the model instabilities and the error can be obtained under strong simplified and somehow unrealistic assumptions on the form of the model dynamics and of the data. For example, if the linear dynamics $\mathbf{M}$, the observation covariance matrix $\mathbf{R}$, and the observation

operator $\mathbf{H}$ are all scalar matrices. With no need of those assumption, and with more generality, the upper bound, Eq. (26), indicates that the MSEF is determined by a convolution of model dynamics and observation error.

In the next sections we will perform numerical experiments under controlled scenarios to investigate the conditions for which the bound holds. In particular, we will study the conditions leading to the smallest possible upper bound, such that the output of a converged DA, *i.e.* its asymptotic MSEF, can be used to infer the LEs spectrum of the model dynamics.

## 215    3    Experimental setting

### 3.1    The Vissio-Lucarini 2020 model

Our test-bed for numerical experiments is the low-order model recently developed by Vissio and Lucarini (2020), hereafter referred to as the VL20. The VL20 model is an extension of the classical Lorenz 96 model (Lorenz, 1996) that includes additional thermodynamic variables. The model is given by the following set of $n$ ODEs (with $n$ being an even integer):

$$\frac{\mathrm{d}X_i}{\mathrm{d}t} = X_{i-1}(X_{i+1} - X_{i-2}) - \alpha\theta_i - \gamma X_i + F \tag{27}$$

$$\frac{\mathrm{d}\theta_i}{\mathrm{d}t} = X_{i+1}\theta_{i+2} - X_{i-1}\theta_{i-2} + \alpha X_i - \gamma\theta_i + G \tag{28}$$

where $X$ represents the momentum, $\theta$ is the thermodynamic variable, and the subscript $1 \leq i \leq n/2$ is the gridpoint index. The model is spatially periodic, and the boundary condition is expressed as:

$$X_{i-n/2} = X_{i+n/2} = X_i \tag{29}$$

$$\theta_{i-n/2} = \theta_{i+n/2} = \theta_i \tag{30}$$

In the VL20 model it is possible to introduce a notion of kinetic energy $K = \sum_{=1}^{n/2} X_j^2/2$ and potential energy $P = \sum_{=1}^{n} X_j^2/2$. Additionally, the model features an energy cycle that allows for the conversion between the kinetic and potential forms and for introducing a notion of efficiency. The parameter $\alpha$ modulates the energy transfer between the two forms, while $\gamma$ controls the energy dissipation rate and $F$ and $G$ are external forcing defining the energy injection into the system.


**Table 1.** Instabilities features of the VL20 model for the three forcing configurations; $n = 36$, and $\alpha = \gamma = 1$.

| (F, G)        | (10, 10) | (10, 0) | (0, 10) |
|---------------|----------|---------|---------|
| $\lambda_1$   | 1.587    | 1.340   | 1.475   |
| $n_0$         | 10       | 7       | 10      |
| $\sigma_{KS}$ | 6.248    | 3.917   | 6.103   |
| $D_{KY}$      | 20.037   | 15.742  | 19.510  |

The model's evolution can be written as the sum of a quasi-symplectic term, which conserves the total energy, and of a gradient term, which describes the impact of forcing and dissipation. In the turbulent regime, the VL20 allows for propagation of signals in the form of wave-like disturbances associated with unstable waves exchanging energy in both potential and kinetic form with the background. In terms of energetics, the difference between the L96 and the VL20 model mirror the one between a one-layer and a two-layer quasi-geostrophic model, because the former features only barotropic processes, while the latter,

features the coupling between dynamical and thermodynamic processes via baroclinic conversion, which makes its dynamics much more complex (Holton and Hakim, 2013). The VL20 model is thus a very good test-bed for research in DA, a further step toward realism from the very successful L96. Further details on the model as well as an extensive analysis of its dynamical and statistical properties can be found in Vissio and Lucarini (2020).

In all the following experiments, we set $n = 36$ implying both model variables $X$ and $\theta$ have 18 components, and consider

three model configurations differing in the values of the external forcings: $F = G = 10$, $F = 10, G = 0$, and $F = 0, G = 10$. Unless otherwise stated, the model runs with the default parameters $\alpha = \gamma = 1$, and it is numerically integrated using the standard 4-th order Runge-Kutta time stepping method with a time step $\Delta t = 0.05$ time units. A summary of the model instability properties with the chosen configurations is given in Tab. 1.

### 3.2 Data assimilation setup

Synthetic observations are generated according to Eq. (2) by sampling a "true" solution of the VL20 model, Eqs. (27), and then adding simulated observational error from the Gaussian distribution $\mathcal{N}(\mathbf{0}, \mathbf{R})$. Observational error is assumed to be spatially uncorrelated thus that the error covariance, $\mathbf{R}$, is a diagonal matrix, and we observe the model components directly, implying that the observation operator is linear and under the form of a matrix, $\mathbf{H} \in \mathbb{R}^{p \times n}$. The observation error variance is set to be 5% of the variance (*i.e.* the squared temporal variability) of the climatology of the corresponding state vector component such

that:

$$diag(\mathbf{R})_i = 5\% Var(X), i = 1, \ldots, \frac{n}{2}; \tag{31}$$

$$diag(\mathbf{R})_i = 5\% Var(\theta), i = \frac{n}{2} + 1, \cdots, n. \tag{32}$$



By linking the observation error to the model variance makes the setup more realistic, but it ties the error amplitude to the choice of the model parameters. For example, the model's state vector variance gets very small when the dissipation is strong, potentially making the $\mathbf{R}$ matrix degenerate. Under such circumstances, the corresponding entries in $\mathbf{R}$ are set to $5 \times 10^{-6}$.

In line with previous studies (*e.g.*, Carrassi et al., 2021), we work with deterministic EnKFs, whereby it is possible to study the filter performance in relation to the model instabilities without the inclusion of additional noise that is inherent to stochastic versions of the EnKFs (Evensen, 2009). In particular we choose to use the finite-size ensemble Kalman filter (EnKF-N, Bocquet et al., 2015) because it automatically computes the required covariance inflation thus saving us from running many inflation tuning experiments. The initial conditions for the ensemble are sampled from the Gaussian distribution $\mathcal{N}(\mathbf{x}_0^t, \mathbf{R})$ with the $\mathbf{x}_0^t$ being the "truth" at $t_0$: this choice signigies that the initial condition error is taken to be equal to the observational error.

The performance of DA experiments will be assessed primarily by using the root mean square error of the analysis, normalised by the observation variance:

$$\text{nRMSEa} = \sqrt{\frac{1}{n}\left(\frac{\sum_{i=1}^{n/2}(X_i - X_{i,truth})^2}{diag(R)_i} + \frac{\sum_{i=1}^{n/2}(\theta_i - \theta_{i,truth})^2}{diag(R)_{i+n/2}}\right)}. \tag{33}$$

The nRMSEa measures the analysis error independent from observation error, allowing for a multivariate assessment of the performance. Unless otherwise stated, observations are taken at every time step, and the experiments last $2,000$ model time units. With this setting, an experiment comprises $40,000$ DA cycles, and when computing time-averages of the nRMSEa, we only consider the last $500$. Finally, and again unless otherwise stated, we shall adopt $N = 40$ ensemble members in the EnKF-N.

## 4 Numerical results

Our analysis focuses on the relation between observational design and filter accuracy, and the relation between the model instabilities and the filter accuracy. By exploiting the novel dynamical-thermodynamical feature of VL20 over its L96 precursor, we will also study the EnKF-N under observational scenarios that alternatively measure the dynamical variable, $\mathbf{X}$, or, the thermodynamical one, $\boldsymbol{\theta}$.

### 4.1 Data assimilation with the VL20 model: general features

Fig. 1 shows the time series of the nRMSEa over the first $100$ time units, for the three main model configurations under consideration. In all cases, the error drops to below $20\%$ of the observational error after approximately $10$ time units (corresponding to $200$ DA cycles), and then fluctuates with oscillations that only sporadically lead the error to exceed $0.3$. The configuration, $(F,G) = (10,0)$ (red line), attains the smaller error, while the other two configurations (blue and green lines respectively) show comparable error levels slightly larger than configuration $(F,G) = (10,0)$. Recall that in the configuration of $(F,G) = (10,0)$, the model is not thermodynamically forced ($G = 0$), and is also slightly stabler than in the other two configurations (*cf.* Tab. 1).

The first connection between the filter performance and the model instabilities is drawn from Fig. 2 that shows the nRMSEa as a function of the number of the ensemble members. In line with previous findings for uncoupled univariate (Bocquet and



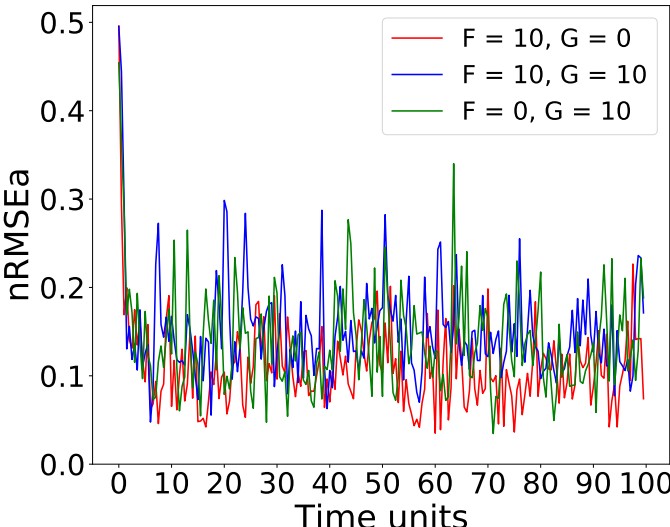

**Figure 1.** Time series of nRMSEa over the first 100 time units ($2,000$ DA cycles) using $\alpha = \gamma = 1$ on $n/2 = 18$ grid points with an ensemble size of $N = 40$.

Carrassi, 2017) and with coupled models (Tondeur et al., 2020), Fig. 2 shows that, even with a multivariate model, the error
converges to very low level as soon as the ensemble size exceeds the number of unstable-neutral modes, $n_0$, and that it does
not further decreases by adding more members. This behaviour is possible because error evolution is bounded to be linear
or weakly nonlinear. This means that one can in principle induce linearity intentionally in the error evolution to meet the
aforementioned relation between filter accuracy and ensemble size and use it to infer the number of unstable-neutral modes.
In a DA experiment, a "practical" way to achieve this is by strengthening the observational constraint (*i.e.*, by increasing the
measurements spatial and temporal density); here we observe the full system's state at every time-step.

The VL20 model represents four main physical mechanisms: i) the transition from kinetic to potential energy; ii) the energy
injection from external forcing; iii) the advection; and iv) the dissipation. Although these processes all participate the evolution
of the model with nonlinear interplay's that cannot be straightforwardly disentangled, we shall try to refer to them when
interpreting the outcome of the DA experiments. In particular, in each experiment we will attempt to identify the prevailing
mechanism over the aforementioned four. We perform three experiments, where we observe the full system state (*i.e.*, $\mathbf{H} = \mathbf{I}_{36}$),
or alternatively $\mathbf{X}$ or $\boldsymbol{\theta}$ alone (implying in both cases $\mathbf{H} \in \mathbb{R}^{18 \times 36}$). Results are given in Fig. 3, that displays the time averaged
nRMSEa (global or for the dynamics or thermodynamics only) over a range of the coefficient $\alpha$ that modulates the energy
transfer rate.

Overall, and as expected, the analysis error is smaller in the observed variables (*cf* the left and mid columns and corre-
sponding color lines), and attains the smallest level when $\mathbf{X}$ and $\boldsymbol{\theta}$ are simultaneously observed (right column). Nevertheless
a few remarkable points can be raised. First, when the system is fully observed, for large $\alpha$ (*i.e.* for large conversion between
available potential energy and kinetic energy) the skills in $\mathbf{X}$ and $\boldsymbol{\theta}$ get very similar (right column): we conjecture this to be


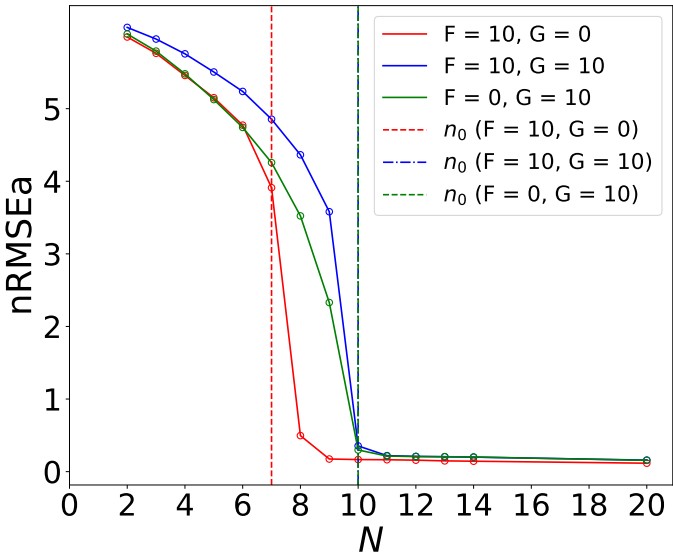

**Figure 2.** The time-averaged nRMSEa for all experiment configurations. The vertical dashed lines indicate the dimension of unstable-neutral subspace, $n_0$. The $n_0$ under the forcing $F = 10, G = 10$ is the same as the forcing condition $F = 0, G = 10$, which shows overlapped vertical lines. For the sake of numerical errors, the neutral mode is chosen as the LE that is closest to 0.

a consequence of the system getting more evenly turbulent with all variables sharing a similar internal variability as energy is exchanged efficiently between the kinetic and potential form. Second, for small $\alpha$ (*i.e.*, small energy conversion), the effect of
external forcing becomes dominant and determines the analysis error of $\mathbf{X}$ and $\theta$ (last column in Fig. 3). For instance, whenever the momentum is externally forced ($F = 10$), the error in $\mathbf{X}$ is systematically smaller than in $\theta$ (first and second rows of the last column): DA is more effective in controlling the dynamics than the thermodynamics even when they are subject to the same observational constraint. The situation is somehow reversed when only the thermodynamics is forced ($F = 0, G = 10$): the analysis error of the momentum and the thermodynamic variable is undifferentiated. With small $\alpha$ and no forcing for the
momentum, the nonlinear momentum advection is limited by the small magnitude of the momentum that is not able to activate much the dynamical variables, so that we observe similar analysis error between the thermodynamics variable and the momentum.

Finally, the effect of the energy transfer and advection can be revealed by looking at the partially observed experiments (left and mid columns). Both mechanisms involve the momentum, making it more efficacious to observe $\mathbf{X}$ than $\theta$ especially in the
energy transfer dominated regime (large $\alpha$). However, in an advection-dominated regimes (small $\alpha$), if $\theta$ is unobserved, $\mathbf{X}$ has limited capability to constrain the error in $\theta$ due to the weak feedback from $\theta$ to $\mathbf{X}$. On the other hand, observing $\theta$ reduces error in $\mathbf{X}$ via the accurate estimate of the advection process of $\theta$ (see mid column).

Further insight on the role of the driving (unstable) variable, and on the interplay between the prevailing physical mechanisms and the analysis error is given by looking at the CLVs (Kuptsov and Parlitz, 2012). In Fig. 4 we show at (normalized time-







**Figure 3.** The nRMSEa with varying energy transfer coefficient $\alpha \in (0, 3)$ (with an interval of 0.1) and dissipation coefficient $\gamma = 1$. The left axis represents nRMSEa while the right axis shows the $\sigma_{KS}$ and the $\lambda_1$ scaled by a factor of 3. The results come from perfect model assumption with observations at each time step where all variables, only $X/\theta$ is observed.

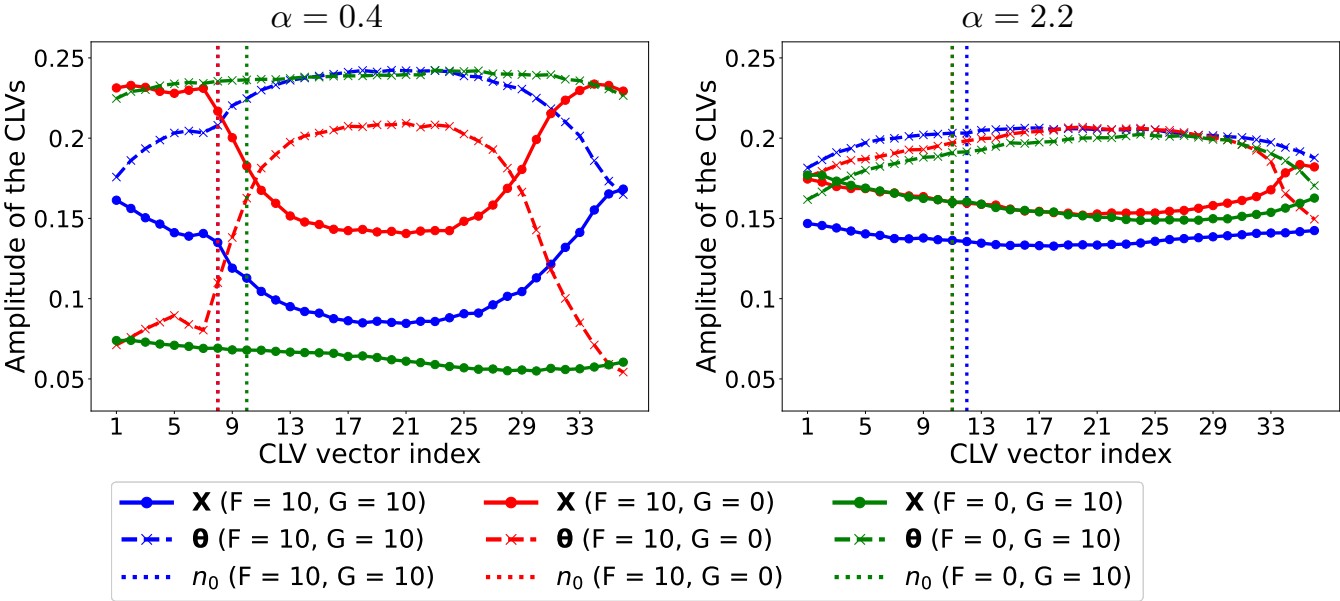

**Figure 4.** Normalized time-averaged amplitude of CLV components (the absolute value) for $\alpha = 0.4$ and $\alpha = 2.2$. In both cases $\gamma = 1$. The vertical lines indicate the corresponding dimension of the unstable-neutral subspace, $n_0$.

averaged) absolute amplitude of CLVs components along the state vector: it tells us which variables/component have the larger influence on each CLVs, thus indicating what processes participate more to a specific direction of error growth/decay.

     As discussed above, the change of $\alpha$ induces the shift from the advection dominated regime to an energy mixing one, where the thermodynamics and the kinetic energy mixes with each other: these two regimes are portrayed in Fig. 4, by selecting $\alpha = 0.4$ and $\alpha = 2.2$. Moreover, these two values of $\alpha$ correspond roughly to those giving the largest differences in nRMSEa

between the momentum and the thermodynamic variables (*cf* left and mid panels of Fig. 3). For small energy exchange ($\alpha = 0.4$ - left column in Fig. 4), the model instabilities are driven by the external forcing, with the driving variable being the one where energy is injected. This is clearly visible when comparing the amplitudes of CLVs between **X** and $\theta$ on the left column of Fig. 4: larger amplitudes of the unstable-neutral CLVs are found in the forced variables. When the momentum and the thermodynamics are equally forced (blue lines), the amplitude of the unstable-neutral CLVs for **X** and $\theta$ are close to each other.

The nonlinear advection process intensifies the error growth, especially for **X**. The nonlinear advection and the momentum is of lesser importance if the momentum is not forced ($F = 0, G = 10$ - green lines) while the thermodynamic processes control dominantly both the stable and unstable subspace. The thermodynamic variable on the stable subspace acts as an energy sink to stabilize the dynamical system. The effect of the thermodynamics is shown noticeably by the large relative amplitude of the CLVs of the thermodynamic variable in the stable subspace when the momentum is directly forced ($F = 10$ - blue and red

line).

     The situation changes sensibly when the energy exchange is the dominant physical mechanism ($\alpha = 2.2$ - right column). This causes a stronger mixing across the model variables so that both **X** and $\theta$ play a comparable role in the unstable-neutral





components of the CLVs leading to similar amplitude of the CLVs for all types of forcing. Remarkably, the effect of the energy conversion also applies to the stable components of the CLVs leading to similar amplitude of the CLVs between $\mathbf{X}$ and $\boldsymbol{\theta}$.

The results in Fig. 4 reveal the effect of the prevailing physical mechanisms on determining the driving unstable variables. The figure suggests what variables should in principle be controlled by targeting measurements on the portion of the system's state vector with larger amplitude on the unstable-neutral CLVs.

Along with $\alpha$, the energy in the system is modulated by the dissipation, $\gamma$: larger values of $\gamma$ implies an efficient removal of energy from the system, and thus reducing the system's variability of both the potential and kinetic energy. At dynamical

level, the parameter $\gamma$ controls the contraction of the phase space as the sum of all Lyapunov exponents (equal to the average flow divergence) is $-n\gamma$. Hence, one expects that larger values of $\gamma$ correspond to weaker instability for the model, as in the case of the classical L96 model (Gallavotti and Lucarini, 2014). Fig. 5 is the same as Fig. 3 but for the dissipation, $\gamma$. Overall, we see that, with large $\gamma$, the system's internal variability reduces and we find similar small errors in both $\mathbf{X}$ and $\boldsymbol{\theta}$. For weaker dissipation, the momentum is better controlled than the thermodynamics. With partial observations (left and

mid columns), the error is much larger than in the corresponding fully observed cases. Similar to Fig. 3, the momentum is generally better reconstructed by the DA than the thermodynamics, although observing the latter appears more efficacious (*i.e.* it leads to smaller analysis error) than observing the momentum. We think that this is due to the prevailing mechanism being the advection of the thermodynamics given that $\alpha = 1$ in these experiments (*cf.* also Fig. 3). The amplitudes of the CLVs along the state vector is studied in Fig. 6. We consider the cases $\gamma = 0.4$ and $\gamma = 1.0$ for which the difference in the nRMSEa between

$\mathbf{X}$ and $\boldsymbol{\theta}$ is roughly the largest (*cf* Fig. 5).

With the leading CLVs strongly affected by the external forcing, the amplitude of the CLVs along the system's components is similar to the pattern of low energy exchange rate in Fig. 4 where $\alpha = 0.4$ even though here, $\alpha = 1$. This confirms that the dynamical regime of our experiments lies in the regime dominated by advection, and dissipation does not mix the kinetic and potential energy diffusely as the energy exchange, but rather it uniformly removes both types of energy without changing the

prevailing physical mechanism. This is also reflected in the consistently low nRMSEa for the observed variable when varying dissipation rates in the partially observed experiments (see Fig. 5). The decreasing analysis error in Fig. 5 corresponds to the increases of $\gamma$, which reduces the dimension of the unstable-neutral subspace with increased relative importance of forced variables in the unstable-neutral subspace as the fast energy removal reduce the amount of energy mixing.

The results of Sect. 4.1 confirm the relation between the performance of DA (in terms of analysis error) and the dimension

and characteristics of the unstable-neutral subspace. In particular, we conclude that successful DA relies on controlling the error in the unstable-neutral subspace by observing the variable that drives the error growth. The VL20 model enabled the investigation of the relation between the DA and the specific physical mechanisms such as the advection, the energy transfer among dynamics and thermodynamics as well as the dissipation. The effect of DA (*i.e.* its efficacy) is strongly influenced by the form of the coupling between the unobserved and the observed variables that is in turn shaped by the prevailing physical

mechanisms.





**Figure 5.** The nRMSEa with varying dissipation coefficients $\gamma \in [0.3, 1.8]$ (with an interval of 0.05) and $\alpha = 1$. The left axis represents nRMSEa while the right axis shows the $\sigma_{KS}$ and $\lambda_1$ scaled by a factor of 3. The results come from perfect model assumption with observations at each time step where all variables, only $X/\theta$ is observed exists.


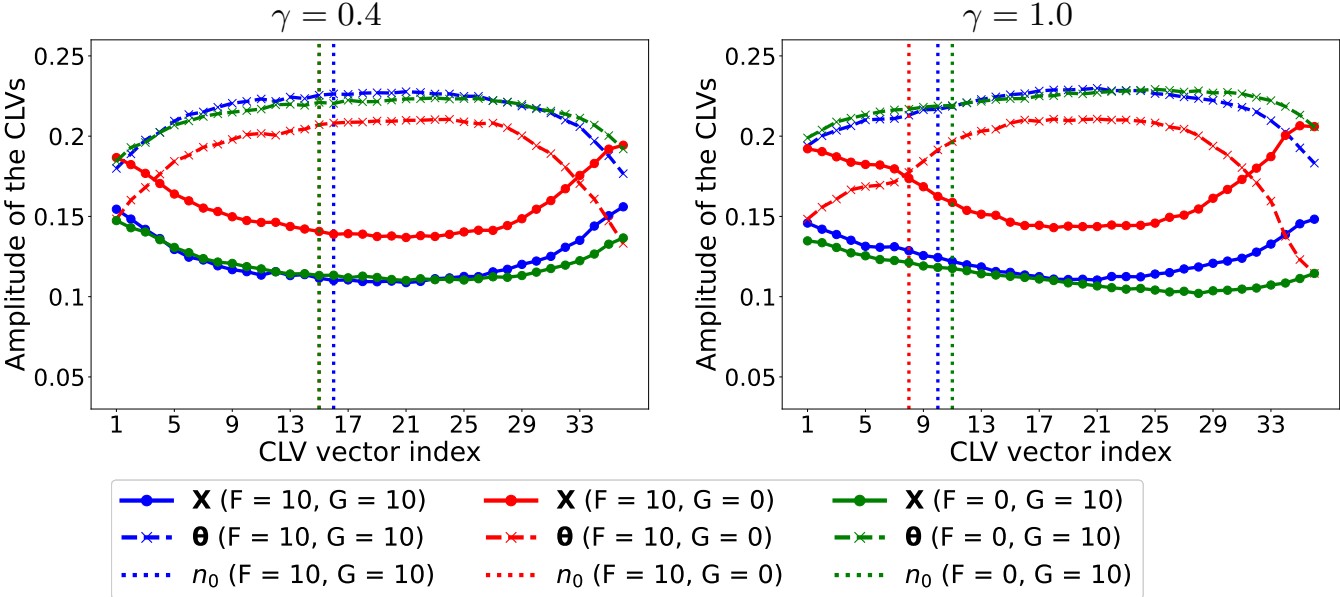

**Figure 6.** Normalized time-averaged amplitude of CLV components for $\gamma = 0.4$ and $\gamma = 1$ with $\alpha = 1$. The vertical lines indicates the dimension of the unstable-neutral subspace.

### 4.2 Inferring the degree of model instability with data assimilation

The derivation in Sect. 2 shows that, in the linear setting, the assimilation error is asymptotically bounded from above by a factor dependent on the observation error, the first LE and the number of unstable-neutral modes of the underlying forecast model. In this section we explore the extent to which this result holds in a nonlinear scenario whereby the observational constraint is strong enough such that the error evolution is maintained approximately linear or weakly-nonlinear. We shall make use of numerical experiments with the VL20 model.

A first insight on the existence of a direct relation between the model instabilities and the skill of the EnKF-N is already provided in Fig. 3 and 5. They display the Kolmogorov-Sinai entropy, $\sigma_{KS}$ (black line) and the first LE, $\lambda_1$ (amplified by a factor of 3 - gray line), along with the nRMSEa (discussed in Sect. 4.1). Even just by visual inspection the figures clearly evidence the high correlation between the analysis error and both the $\sigma_{KS}$ and $\lambda_1$.

The nature of this relation is further studied in Fig. 7 that shows scatter plots between the nRMSEa (with black markers) and $\sigma_{KS}/\lambda_1$ in a log-log scale. Points are relative to experiments the forcing values given in the panels' legends and with varying energy exchange and dissipation rates in the range $(\alpha \times \gamma) \in [0.1, 3) \times [0.3, 1.8)$. Here, the EnKF-N assimilates the full state vector at each time step. The analysis error appears in a linear relationship with either $\sigma_{KS}$ or $\lambda_1$, as long as $\ln(\text{nRMSEa}) \geq -4$. The existence of such a quasi-linear relationship provides the possibility to infer $\sigma_{KS}$ and/or $\lambda_1$ based on the outcome of DA.


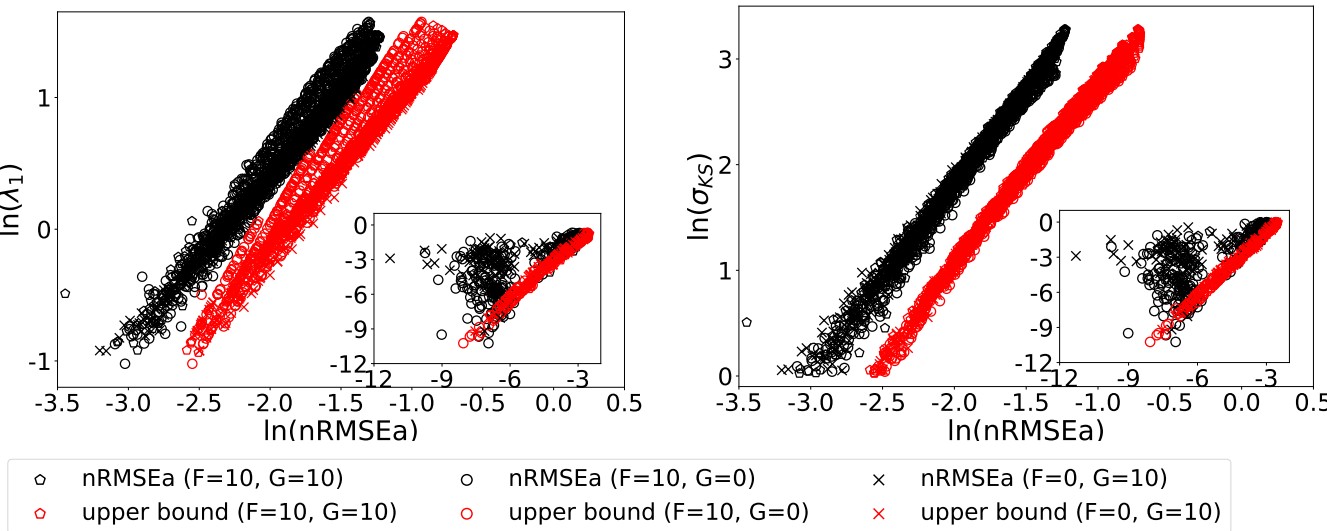

**Figure 7.** Scatter plots of $\lambda_1$ (left) and $\sigma_{KS}$ (right) against the nRMSEa for experiments with observations of the entire state vector at each time step. The theoretical analysis error upper bound are also displayed (red markers). The log scale is used on both axes. The experiments use the model forcing given in the legend and the energy rate and dissipation in the range $(\alpha \times \gamma) \in [0.1, 3) \times [0.3, 1.8)$ with an interval of $0.1 \times 0.05$. The stable configurations ($\sigma_{KS} < 1$) have been excluded. The inset shows the weakly unstable model configurations ($0 < \sigma_{KS} \leq 1$).

The scatter plots also demonstrate the validity of the upper bound (red markers) of Eq. (26) in Sect. 2. To compute the bound we set the coefficient related to observation, $\beta = 1$, as it is compared to analysis errors that normalized by observational error. The nRMSEa is bounded by the theoretical upper bounds for most of the model configurations considered. The spread of upper

bounds points for given $\lambda_1$ (left panel) reflect the various values of $n_0$ under similar $\lambda_1$. The better correspondence (narrower spread of the scattered points) in the plane $nRMSEa$ with $\sigma_{KS}$ (right panel) shows the importance of including both the dominant error growth rate, $\lambda_1$, and the unstable-subspace dimension, $n_0$, - both present in $\sigma_{KS}$ - to better characterise the system's instabilities. The correspondence between $\sigma_{KS}$ and the theoretical upper bound could also be a result of the relation between $\lambda_1$ and $\sigma_{KS}$ as in highly turbulent case, there is a linear relation between $\lambda_1$ and $\sigma_{KS}$ in (Gallavotti and Lucarini,

395    2014).

The linear relation does not hold when $\ln(\text{nRMSEa}) < -4$ (see the black markers distribution in the panels' inset). We explain this behaviour in the following way. The wide clouds of points in correspond all to model configurations with very small $\sigma_{KS}$ and $\lambda_1$. In these quasi-stable dynamics, the error growth in between successive analysis is very little, with occasional error decay. The observational error, which is random and white-in-time, will be often larger than the forecast error and will

dominate the analysis error, thus breaking its direct dependence on the instability-driven forecast error. In addition, in the weakly unstable model configurations, the most unstable direction behaves almost like a neutral mode which also breaks the


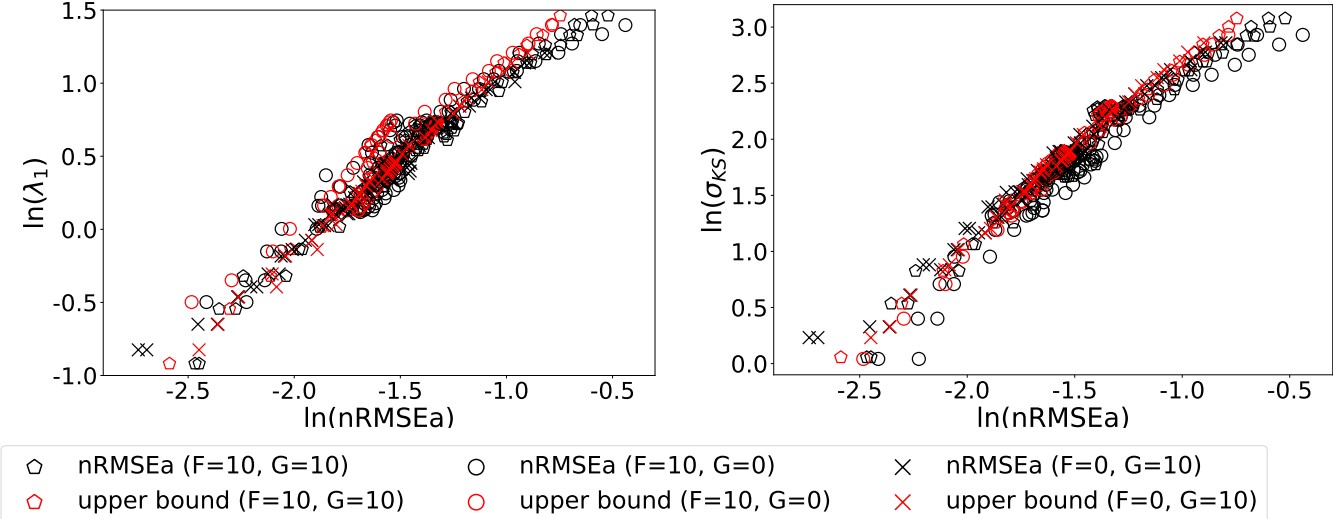

**Figure 8.** Scatter of $\sigma_{KS}$ (left) and $\lambda_1$ (right) against nRMSEa for experiments where either the momentum or the thermodynamic variable alone is observed. The log-log scale is used in both axes, and points represent experiments with the same model parameter values used in Fig. 3 and 5, excluding weakly unstable cases with $\sigma_{KS} < 1$, and hence excluding cases where $\ln(\text{nRMSEa}) < -4$ similar to Fig. 7.

assumptions of the theoretical upper bound for unstable models. In fact, we do not show the weakly unstable results with $\sigma_{KS} < 1$ for both $\sigma_{KS}$ and $\lambda_1$.

The error bounds in Sect. 2 relies on the assumption of linear error evolution, a condition that we met in our experiments
thanks to a strong observational constraint, with (synthetic) measurements covering the full state vector at each time-steps. These conditions are rarely achievable in practice, so it is relevant to explore how results will change with lighter observational constraint. There are three direct ways to achieve this by acting on (i) the number/type of measurements, (ii) the measurement error, and/or, (iii) the temporal frequency.

The effect of the first is studied in Fig. 8 that is similar to Fig. 7 but for DA experiments whereby only one of each variable
in the VL20 is observed.

The impact of partially observing the system causes the emergence of a weakly quadratic relationship between the analysis error and either $\sigma_{KS}$ or $\lambda_1$. However, the analysis error is still uniquely and monotonically related to them especially for $\sigma_{KS}$. A quadratic law requires one additional coefficient to be determined compared to a linear law, yet the mere existence of such a law suggests again that one could in principle infer $\sigma_{KS}$ and/or $\lambda_1$ based on the analysis error. With the relaxed observation
constraint, the analysis error can (and indeed do so in several instances) exceed the theoretical upper bound. However, the general trend of the numerical experiments still follow the theoretical upper bound.

We study the effect of changing the amplitude of the observational error in Fig. 9. Results reveal that varying the observation error in the range of $5\% - 10\%$ does not break the quasi-linear relationship between the analysis error and $\sigma_{KS}$ or $\lambda_1$. The nRMSEa is quite insensitive to the observation variance due to the normalization. Nevertheless, the upper bound is not violated


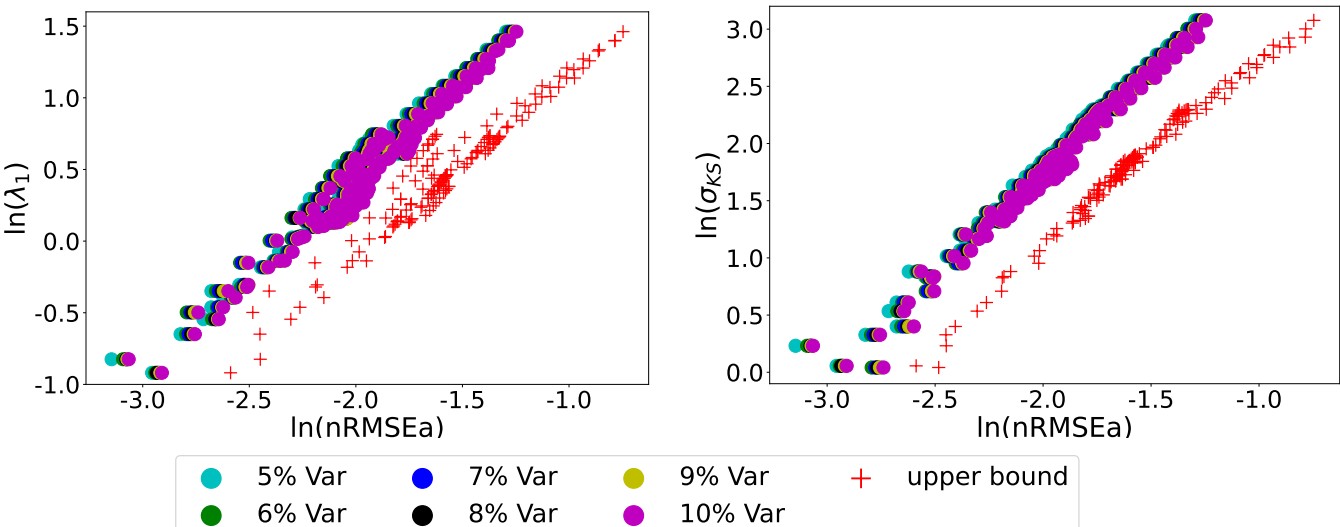

**Figure 9.** Scatter plots of $\sigma_{KS}$ (left) and $\lambda_1$ (right) against nRMSEa. The log-log scale is used on both axes. The different points refer to experiments with different observation error given in the legend and model parameter as in Fig. 3 and 5 and excluding weakly unstable cases with $\sigma_{KS} < 1$, and hence excuding cases where $\ln(\text{nRMSEa}) < -4$ similar to Fig. 7.

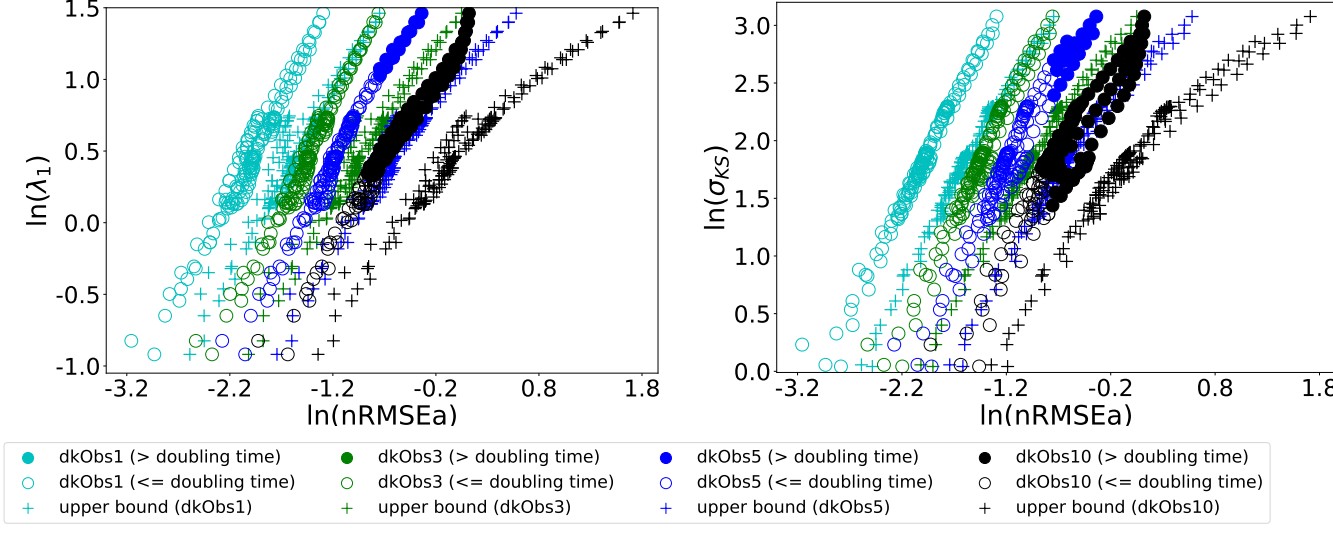

**Figure 10.** Scatter plots of $\sigma_{KS}$ (left) and $\lambda_1$ (right) against nRMSEa. The log-log scale is used on both axes. The different points refer to experiments with different observation interval given in the legend and model parameter as in Fig. 3 and 5 and excluding cases when $\sigma_{KS} < 1$. The filled circle represents the cases where the observation interval exceeds the doubling time of the error.

as in Fig. 7 and the slope of the nRMSEa from the numerical experiment is remarkably similar to the slope of the theoretical bound.





Finally, the impact of varying the observation frequency is explored in Fig. 10. It is patent that decreasing the frequency leads to blurring the linear relation between the analysis error and the $\sigma_{KS}$ or $\lambda_1$. There is a clear deviation from the trend of the theoretical upper bound and from the uniqueness of the relation between analysis error and $\sigma_{KS}$, as soon as the observational time interval exceeds the error doubling time (that is inverse related to $\lambda_1$), and DA error evolves beyond the linear regime. However, for frequent enough observations a linear relation similar to the upper bound appears and, again, one could in principle deduce $\sigma_{KS}$ and/or $\lambda_1$ based on DA.

Finally note that, the different effect of the observational noise and data frequency depicted in Fig. 9 and Fig. 10 is consistent with the findings in Bocquet and Carrassi (2017) (their appendix) where it is shown how observation frequency is a much more effective driver to induce (or to break) linear error evolution.

## 5   Conclusions

It is sometimes of great importance to be able to obtain information on the instability of a system of interest by performing data analysis of suitably defined observables. This is of key importance when one does not have direct access to the evolution equations of the system or when the analysis of its tangent space is too computationally burdensome. As an example, quantitative information on the degree of instability of a chaotic system can be extracted using extreme value theory by studying the statistics of close dynamical recurrences as well as of extremes of so-called physical observables (Lucarini et al., 2014, 2016). The use of such a strategy has shown a great potential for the analysis of geophysical fluid dynamical models in a highly turbulent regime (Gálfi et al., 2017) as well for the understanding of the properties of the actual atmosphere (Faranda et al., 2017; Messori et al., 2017).

In this study, we have addressed this problem by taking the angle of DA. The relation between DA and the instability of the dynamical system where it is applied has long been studied (see *e.g.* Miller et al., 1994; Carrassi et al., 2008), and has been used to design DA techniques in various field of geosciences (Carrassi et al., 2021; Albarakati et al., 2021). Here, we have reversed this viewpoint and investigated the possibility of using DA to infer fundamental quantities of the underlying dynamics, in particular the Lyapunov exponents, $\lambda_i$, or the Kolmogorov-Sinai entropy ($\sigma_{KS}$). The basic idea is to look at DA as a control problem, and relate our ability to control the system, *ceteris paribus*, to its underlying instability. We have leveraged on a stream of previous works that set the theoretical foundation and that proved the convergence of the error covariance of the Kalman filters onto the unstable-neutral subspace of the dynamical system. Based on this, we derived here an upper bound of the Kalman filter forecast error, *i.e.* under the assumptions of a linear model dynamics and a linear observation operator. The upper bound is very informative as it relates the error's amplitude to all of the essential descriptors of the model instabilities on the one hand and of the DA on the other. These are the dimension of the unstable-neutral subspace, $n_0$, the first Lyapunov exponent, $\lambda_1$, the frequency of the observation assimilation, $\Delta t$, and the observation error, $\beta_k$. By properly normalising the bound by the observation error, it can be written as a function of the model dynamical properties exclusively.

The existence of a relation between $\lambda_1$ or $\sigma_{KS}$ and the DA skill, as well as the validity of the bound, has then been investigated in a nonlinear scenario using numerical experiments. We have used the EnKF-N (Bocquet et al., 2015) as a prototype of





deterministic EnKF (Evensen, 2009) and the new model developed by Vissio and Lucarini (2020). The VL20 is an extension of the widely used Lorenz 96 model that includes a thermodynamic component. While maintaining all of the virtues of a low-dimensional model suitable for investigations on new methods at low computational cost, VL20 is conceptually much richer than the original L96 model. In particular it allows for the exchange of energy between a kinetic and potential form, which, together with forcing and dissipation, provides the fundamental framework for the Lorenz (1955) energy cycle. Additionally,

as advection impacts temperature-like variables, one can observe the emergence of more complex dynamical behaviors. By changing the value of its key parameters, and in particular of those determining its forcing and dissipation, the model explores various dynamical regimes, ranging from fixed point, periodic, quasi-periodic, and chaotic behaviour. In terms of DA, the VL20 model has the attractive feature that it includes two qualitatively different set of variables, associated with dynamics and thermodynamics, respectively. Hence, it is possible to explore the problem of having partial observation beyond focusing of

the spatial extent of the observations only.

We demonstrate that the skill of the EnKF-N is directly linked to both $\lambda_1$ and $\sigma_{KS}$. Whenever the error within the EnKF-N cycles are kept sufficiently linear via a strong observational constraint, the relation is clearly linear too. By relaxing the observational constraint (by either reducing the frequency of measurements or by increasing their noise) deviation from linearity emerge. Nevertheless, the linear relation is very robust against the level of observational noise (within certain range) while it

turns quadratic once the interval between successive measurements gets too large and it exceeds the system's doubling time. Similarly, we found out that the theoretical upper bounds for the errors, derived for linear system, still holds as long as the observational constraint is strong enough, but are then violated.

The error bound and the linear relation between error and $\lambda_1$ and $\sigma_{KS}$ represent a potentially powerful direct way to infer $\lambda_1$ and $\sigma_{KS}$ by looking at the output of a DA exercise. Knowing the analysis error (or a suitable approximate estimate of it)

computing $\sigma_{KS}$ or $\lambda_1$ requires the unstable-neutral subspace dimension, $n_0$. It can be obtained by looking at the analysis error convergence when increasing the ensemble size, $N$: $n_0$ will be equal to $N^* - 1$ where $N^*$ being the smallest ensemble size for which the error reaches is minimum.

There are some follow up questions that emerge naturally from this work. Among those, we are currently considering how these results will change when performing DA for state and parameter estimation. In this context, a relevant recent study has

shown how the minimum number of ensemble members, $N^*$, will need to be increased to include as many members as the number of parameters to be estimated (Bocquet et al., 2020). By modifying its parameters, the model's instabilities properties will change too, potentially inducing a catastrophic change (a tipping point) of its long term behavior. Data assimilation will then need to infer the best parameter values to track the data signal and keep the DA solution on its same region of the bifurcation diagram.

*Code availability.*    The Python script for the plotting and data assimilation experiments is available at https://github.com/yumengch/InferDynPaper, which also is dependent on version 1.1.0 of the Python package DAPPER (https://github.com/nansencenter/DAPPER/releases/tag/v1.1.0).



*Author contributions.* Yumeng Chen designed and conducted the experiments, and prepared the manuscript. Alberto Carrassi and Valerio Lucarini both provide the original idea and the writing of the manuscript. All authors have then contributed to develop the work.

*Competing interests.* Alberto Carassi and Valerio Lucarini are editor of NPG.

*Acknowledgements.* The authors are thankful to Patrick Raanes (NORCE, NO) for his support on the use of the data assimilation python platform DAPPER. YC and AC have been funded by the UK Natural Environment Research Council award NCEO02004. VL acknowledges the support received from the EPSRC project EP/T018178/1 and from the EU Horizon 2020 project TiPES (grant no. 820970).





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
