# Peer review of "Inferring the instability of a dynamical system from the skill of data assimilation exercises"

_Nonlinear Processes in Geophysics, 2021_

## Author Comment (AC1)

**Responses to Reviewers**

We thank both Reviewers for their careful reading of our manuscripts and the very valuable suggestions for improvements. We have done our best to address all of the Reviewers' remarks and provide a detailed point-by-point responses in the following. In the interested of readability, we report each of the Reviewers' remark followed by our answers.

We hope the current version of the manuscript meets the high standard expected for publication on NPG.

**Reviewer 1**

This paper explores an interesting area of using the DA method to infer some basic properties of the dynamic system. The hypothesis is investigated on the Vissio-Lucarini 2020 model. The method is sound. However, the authors should address the below comments.

1. Generally, the length of this manuscript is a bit too long. The authors need to revise and strengthen the direction of their work. Either has a clear aim and objectives or poses the main research question with sub-questions. At the moment your introduction has three parts: "Lyapunov vectors and related measures of chaos in a nutshell", "Data assimilation in chaotic systems: the signature and the use of chaos" and "This paper: data assimilation as a tool to interrogate the dynamics". Besides a literature review and introduction of these three topics, the authors should also explicitly state the correlations of these three topics and connect them smoothly. And finally, have aim and objectives (or research questions) drive an overall high-level methodology for the paper.

   **Answer:** Thank you for the suggestion. We agree with the reviewer that the introduction is particularly long. We shortened paragraphs discussing the local variability of dynamical systems, and the particle filters. However, because our manuscript combines DA, dynamical systems, and predictability, we believe that the remaining introduction, though still long, is required to present these topics for potential readers from various background.

   To address the lack of correlations of these topics and the clarity of the objectives of the manuscript, we added sentences to connect each topic, and strengthened the main question of the paper in Section 1.3. For example, the title of Section 1.3 now is "This paper: can data assimilation be used to reconstruct the dynamical properties of the system?". Additionally, we have added the following paragraph:

   "A major but not exclusive issue is that LEs estimation' algorithms require computing the tangent space of the dynamical system, a task usually unfeasible for high-dimensional systems, or impossible when the model equations of are not explicitly accessible. On the other hand, the existence of a relationship between the DA and the unstable-neutral subspace suggests reversing the view-angle: use DA as a tool to estimate the properties of a given system that would be otherwise

very difficult to compute. As a model agnostic technique, DA, and in particular ensemble-based methods such as the EnKF, can be applied to any model without the need of computing the tangent space. This makes the EnKF a potentially powerful instrument to reveal the stability properties of a dynamical system. This is the goal of this work. Specifically, we shall investigate whether we can use DA to infer the spectrum of the LEs and the Kolmogorov-Sinai entropy ($\sigma_{KS}$) of the system whereby data are assimilated. "

2. In section 4.2, the authors considered relaxing the observational constraint and concluded the linear relation is very robust against the level of observational noise (within a certain range) while it turns quadratic once the interval between successive measurements gets too large. Could the authors also provide some insights into these results? Could you explain the potential reasons for the rather less effect of the observational noise and rather a large effect of the observation frequency?

   **Answer:** We thank the Reviewer to remark the lack of sufficient explanation. The effectiveness of the observational interval, as opposed to the observational noise, is a due to the former being directly related to the magnitude of the non-linear advection term in the model. This is explained in detail in Bocquet and Carrassi (2017). We also added a mathematical explanation for the basis of the linear relationship for the upper bound, which requires frequent observations. Longer observations breaks the condition of such linear relationship. We have modified the text in section 4.2 (line 388) of the new version of the manuscript as follows:

   *"The linear relationship of the upper bound can be explained by its formulation in Eq. 26, where the exponent $e^{2\lambda_1 \Delta t} - 1$ can be approximated as $2\lambda_1 \Delta t$ if $2\lambda_1 \Delta t$ is sufficiently small."*

   and in line 428 of Section 4.2

   *"The larger sensitivity to the observation frequency than to observation noise (cf Fig. 9 and Fig. 10) is a direct consequence of the different effects these two factors have in determining the degree of non-linearity of the error. This is explained, for the L96 model, in the Appendix of Bocquet and Carrassi (2017) using a dimensional analysis. The key point is that the observation frequency modulates directly the magnitude of the nonlinear term in the model, namely the advection. Decreasing the observation noise, while effectively reducing the analysis error, is not sufficient to keep the error dynamics linear if $\Delta t > 0$ and the model is chaotic."*

**Reviewer 2**

In this article, the authors present a theoretical upper bound for the analysis error of a chaotic dynamical system under perfect and linear model conditions. This upper bound considers the properties of the observation network (Spatio-temporal distribution of the observations) as well as the dynamical properties of the system. Then, the

authors evaluate the relation between analysis errors and different observing networks and dynamical properties using an extension of the Lorenz 96 system. The results are quite interesting and show the impact of observing different variable types and changes in the system's dynamics. The authors also show that the proposed upper bound for the analysis error holds for this system under weakly non-linear regimes and that the magnitude of the analysis error is linked to fundamental dynamical properties such as the leading Lyapunov exponent or the Kolmogorov-Sinai entropy.

The paper is well written. The discussion of the motivation, methodology, and results is clear. I have only some minor comments and questions for the authors.

1. It would be interesting to add a discussion about how these results could change in the presence of model error. At least some hypotheses (like the convergence of $\mathbf{P}_f$ to the unstable subspace) may not hold in this case. Some discussion is included about the parametric model error, but also structural model errors are important.

   **Answer:** We thank the Reviewer for raising this important point. There is not a clear picture in the case of structural model error, whereby the general model error is intended as per a model following a dynamics different from the one derived from the observations (and therefore with a generally different phase space). However, this type of complex, albeit unavoidable, model error is often and conveniently treated as additive noise. In this case, the convergence of the Kalman filter error covariance, $\mathbf{P}^f$, onto the unstable subspace has been studied in Grudzien et al. (2018a) and Grudzien et al. (2018b). We briefly recalled these results in Sect. 1.2 of the original version of the manuscript. The essential point is that, for hyperbolic dynamics, with stochastic model noise, the error covariance will no longer be fully confined within the unstable-neutral subspace, but will have probability one to project everywhere and thus also on the stable modes. By doing so, the error along those directions, while being continuously dampen by the dynamics is also constantly kept alive by the injection of noise.

   While this remains to be investigated, we argue that the existence of a clear monotonic relationship between analysis error and $\lambda_1$ will still hold in the presence of model error. The relation to $\sigma_{KS}$ might also still stand because the correction would come from weakly stable modes. However, the conjecture need to be validated by numerical experiments that are out of the scope of this manuscript.

   We have modified the text as follows in line 495:

   *"The linear relation between error and $\lambda_1$ and $\sigma_{KS}$ will certainly be more complicated with model errors. From Grudzien et al. (2018a) and Grudzien et al. (2018b) we know that the KF error covariance will no longer be fully confined within the unstable-neutral subspace, but will maintain projections with probability one everywhere, and thus also on the stable modes. Those projections would be asymptotically zero in the absence of model noise. While this remains to be investigated, we argue that the existence of a clear monotonic relationship between analysis error and $\lambda_1$ will still hold in the presence of model error. The relation to $\sigma_{KS}$ might also still stand because the correction would come from*

*weakly stable modes. However, the conjecture need to be validated by numerical experiments that are out of the scope of this manuscript."*

2. L473 In this paragraph an idea on how to use DA to estimate $\lambda_1$ or $\sigma_{KS}$ is presented. Results suggest that this is possible but requires investigating the behavior of the system under different dynamics. Can we estimate $\lambda_1$ if we have only one DA system with a particular observing network (like in operational DA)? Can the relations obtained for this particular system be extended to other systems?

L474 It is stated that using the analysis error and $n_0$, an estimate of $\lambda_1$ can be obtained. This is unclear for me. The relation between $\lambda_1$ and the analysis error seems to be empirically obtained in this paper. Are the authors assuming that the analysis error is equal to its upper bound (which is theoretically linked with $\lambda_1$ and $n_0$)?

**Answer:** We thank the reviewer for raising the issues. We believe these issues arise from the lack of clarity of the manuscript. We hope our added content can amend the issue.

The estimate of $\lambda_1$ depends on $n_0$, and the existence of the quasi-linear relation between the DA skill and $\lambda_1$ in the log-log space. The observing network should be able to provide these information, which requires frequent observations. One of the benefits of our proposed approach is to efficiently estimate $\lambda_1$ or $\sigma_{KS}$ when the model parameter changes.

In the revised manuscript, we further clarify two approaches to estimate $\lambda_1$ or $\sigma_{KS}$. One approach is based on the upper bound, the other on the linear relation between the nRMSE and the dynamical properties. Details of our description can be found in the conclusion (line 477 Sect. 5) stated as follows:

*"The error bound and the linear/quasi-linear relation between the error and $\lambda_1$ or $\sigma_{KS}$ represent two direct ways to infer $\lambda_1$ and $\sigma_{KS}$ by looking at the output of a DA exercise. First, we can use the bound (Eq. 26.) to estimate $\lambda_1$ for a specific dynamical model, based on (normalised) error output of a DA exercise. This requires the unstable-neutral subspace dimension, $n_0$, that can be obtained , in the case of EnKF-like methods, by looking at the analysis error convergence for increasing ensemble size, N: $n_0$ will be equal to $N^* - 1$ where $N^*$ being the smallest ensemble size for which the error reaches is minimum. This procedure will give us an underestimate of $\lambda_1$. Nevertheless, our results (cf Fig. 7) seem to suggest that the amount of the underestimation is small and, notably, constant across a range of different model configuration (and thus possibly quantifiable).*

*Our numerical experiments indicate a second way to estimate $\lambda_1$ or $\sigma_{KS}$ from the skill of DA. The linear/quasi-linear relationship between normalised DA error and $\lambda_1$ or $\sigma_{KS}$ (cf Fig. 7) exists for both the derived upper bound in Eq. 26 and numerical experiments, and is tested under various observation constraint. The existence of the relationship for the upper bound implies that the relationship may exist for other dynamical systems as long as the time between analysis $\Delta t$ is sufficiently frequent because the upper bound is based on the assumption*

*of a (quasi-)linear model. To utilize the relationship for a specific dynamical system, a few DA experiments using different set of parameters of the dynamical system is required. A linear relation can be obtained by linear regression from the selected data, by which a relatively accurate $\lambda_1$ or $\sigma_{KS}$ for other parameters of the dynamical system can be inferred. Unavoidably, for the selected set of parameters, the method requires the $\lambda_1$ or $\sigma_{KS}$ to be known, which possibly can be obtained by computational methods such as the one proposed by Wolfe and Samelson (2007). However, the resulting linear relation can lead to a computationally efficient approach for other sets of parameters of the dynamical system with an estimate more accurate than the one from the upper bound in Eq. 26.".*

3. L26 Kolmogorov-Sinai entropy (or metric)

   **Answer:** Corrected in line 28 with "Kolmogorov-Sinai entropy (or metric entropy)".

4. L26 and can be identified as?

   **Answer:** Corrected in line 30 as "can be estimated as".

5. L83 asymptotic unstable-neutral modes?

   **Answer:** The sentence is removed.

6. Equation 7, please check the correctness of this equation.

   **Answer:** The equation is correct. The equation follows from Bocquet and Carrassi (2017) to ensure a positive-definite information matrix, and we add an explanation for it in line 146:

   "which implies that each term of the information matrix should be positive-definite,"

7. L227 Please revise the definition of the potential energy (the summation index and the definition that should include $\theta$ )

   **Answer:** Thank you. Corrected in line 226.

8. L261 signifies

   **Answer:** Thank you. Corrected in line 260.

9. Figure 1 Does this figure corresponds to the fully observed case?

   **Answer:** Yes, and the information is added to the caption:

   "Time series of nRMSEa over the first 100 time units ($2,000$ DA cycles) using $\alpha = \gamma = 1$ on $n/2 = 18$ grid points with an ensemble size of $N = 40$ with the entire state vector observed at every time step."

10. L267 The last 500 DA cycles or model time units?

    **Answer:** Thank you. We added this information (500 time units) in line 267.

11. L285 very low values?

    **Answer:** We replace low level to low values in line 284 now.

12. L292 Please revise the sentence starting with "Although these processes ..."

    **Answer:** Thank you. We rephrased the sentence in line 291:

    "Although these processes all participate the evolution of the model, the nonlinear interplay cannot be straightforwardly disentangled. ".

13. Caption Figure 3: ... where all variables, or only X, or only $\theta$ are observed? Also describing each color line in the caption would be better. Also for Figure 5.

    **Answer:** Done. We added:

    "... where all variables, only **X** or only $\theta$ is observed. The blue dashed line indicates the nRMSE of the **X** variable, the dashed red line represents the nRMSE of the $\theta$ variable, and the dashed green line shows the nRMSE of the entire state vector. "

14. L319 The description of Figure 4 is unclear. Variable can refer to a grid point or different variable types. Maybe better to say variable type instead of just variable.

    **Answer:** Done. We write "variable type" instead of variable in line 319.

15. Figure 4: Are the variables normalized before the respective CLV amplitude is computed? Results, in this case, are reasonable, but I wonder how this analysis can be extended to more complex systems with variables with different ranges of variability and possibly different units.

    **Answer:**

    Thank you for the question. To make variables comparable we had to normalise them. We did so by using the CLVs total amplitude. When working with more realistic scenarios where variables are also expressed in different units with possible very different ranges, the aforementioned normalisation needs to follow an adimensionalisation.

16. Figure 7: In the caption, it is not clear if $\sigma_{KS}$ represents the stable configurations or the weakly unstable model configurations.

    **Answer:** We corrected a mistake in the caption. $\sigma_{KS} < 0$ should be stable configurations.

17. L388 errors normalized?

    **Answer:** Corrected.

**References**

Bocquet, M. and Carrassi, A.: Four-dimensional ensemble variational data assimilation and the unstable subspace, Tellus A: Dynamic Meteorology and Oceanography, 69, 1304 504, 2017.

Grudzien, C., Carrassi, A., and Bocquet, M.: Asymptotic forecast uncertainty and the unstable subspace in the presence of additive model error, SIAM/ASA Journal on Uncertainty Quantification, 6, 1335–1363, 2018a.

Grudzien, C., Carrassi, A., and Bocquet, M.: Chaotic dynamics and the role of covariance inflation for reduced rank Kalman filters with model error, Nonlinear Processes in Geophysics, 25, 633–648, 2018b.

Wolfe, C. L. and Samelson, R. M.: An efficient method for recovering Lyapunov vectors from singular vectors, Tellus A: Dynamic Meteorology and Oceanography, 59, 355–366, https://doi.org/10.1111/j.1600-0870.2007.00234.x, 2007.